# Virus-mediated export of chromosomal DNA in plants

Marco Catoni [1,2], Emanuela Noris [3], Anna Maria Vaira [3], Thomas Jonesman [1], Slavica Matić[3], Reihaneh Soleimani[4,5], Seyed Ali Akbar Behjatnia [4], Nestor Vinals[3], Jerzy Paszkowski [1] & Gian Paolo Accotto [3]

The propensity of viruses to acquire genetic material from relatives and possibly from infected hosts makes them excellent candidates as vectors for horizontal gene transfer. However, virus-mediated acquisition of host genetic material, as deduced from historical events, appears to be rare. Here, we report spontaneous and surprisingly efficient generation of hybrid virus/host DNA molecules in the form of minicircles during infection of *Beta vulgaris* by *Beet curly top Iran virus* (BCTIV), a single-stranded DNA virus. The hybrid minicircles replicate, become encapsidated into viral particles, and spread systemically throughout infected plants in parallel with the viral infection. Importantly, when co-infected with BCTIV, *B. vulgaris* DNA captured in minicircles replicates and is transcribed in other plant species that are sensitive to BCTIV infection. Thus, we have likely documented in real time the initial steps of a possible path of virus-mediated horizontal transfer of chromosomal DNA between plant species.

[1] The Sainsbury Laboratory, University of Cambridge, Cambridge CB2 1LR, UK. [2] School of Biosciences, University of Birmingham, Birmingham B15 2TT, UK. [3] Institute for Sustainable Plant Protection, National Research Council of Italy, Torino 10135, Italy. [4] Plant Virology Research Center, College of Agriculture, Shiraz University, Shiraz 71441-65186, Iran. [5]Present address: Department of Plant Protection, Isfahan (Khorasgan) Branch, Islamic Azad University, Isfahan 81595-158, Iran. These authors contributed equally: Marco Catoni, Emanuela Noris. Correspondence and requests for materials should be addressed to G.P.A. (email: gianpaolo.accotto@ipsp.cnr.it)

The acquisition of genetic material originating from unrelated species, also known as horizontal gene transfer (HGT), is of primary importance in the evolution of genomes. Although the majority of HGT events between eukaryotes have been revealed by computational searching and seem to involve transposable elements (TEs), the mechanisms and frequencies of transfer between species are still in dispute[1]. Furthermore, the vectors involved in these apparently sporadic events of eukaryotic HGT remain largely unknown[2,3].

Viruses are potential vectors for HGT as they have a high propensity for recombination, their genetic material penetrates host cells, and they are efficiently transmitted between hosts[4]. Thanks to the increasing number of available host and viral genome sequences and to paleo-virology studies, many examples of potential gene flow between viral and host genomes in various eukaryotic lineages have been proposed, both "virus-to-host" and "host-to-virus"[4,5]. The most intriguing example is that of baculoviruses, which occasionally acquire DNA fragments from infected hosts into their relatively large genomes, predominantly those encoding TEs[6,7]. In addition, single-strand RNA viruses can potentially encapsidate host RNAs into their virions. This has been observed in insects[8] and plants[9,10]. Therefore, it has been suggested that viruses are HGT vectors between eukaryotes that share the same viral pathogens. However, to date, virus-mediated HGT has not been examined experimentally or monitored in real time.

Here, we describe spontaneous and efficient formation of novel hybrid DNA minicircles composed of viral and plant genomic DNA sequences. They arise in sugar beet plants (*Beta vulgaris*) infected by a circular single-stranded DNA virus, *Beet curly top Iran virus* (BCTIV), belonging to the *Geminiviridae* family. The viral portion of the hybrid minicircles always encompasses regulatory regions crucial for virus life cycle. Therefore, minicircles replication and encapsidation are supported *in trans* by BCTIV. We demonstrate that the hybrid minicircles can replicate, systemically spread and produce RNA transcripts in other plant hosts infected with BCTIV. Our results reveal that in the host/virus combinations tested here, efficient and reproducible recombination between viral and host genomes can be observed in real time, suggesting that ssDNA viruses have a high potential to act as vectors for HGT between host plants. Interestingly, efficient formation of minicircles seems to be species specific, determining the direction of information flow between virus hosts.

## Results

### Identification of hybrid viral/host DNA minicircles in BCTIV-infected sugar beet fields.
To identify the infectious agent(s) causing curly top disease of sugar beet in Iran, DNA was extracted from diseased plants and analyzed. Considering the symptoms, the pathogen was suspected to be *Beet curly top Iran virus* (BCTIV), a geminivirus with a circular 2845-nt genome containing a single *Eco*RI restriction site[11]. Rolling circle amplification (RCA) of the extracted DNA and digestion with *Eco*RI revealed the expected 2845-nt fragment. Surprisingly, we also detected fragments of 1300–1600 nt (Fig. 1a) that suggested the propagation of DNA molecules smaller than the viral genome in BCTIV-infected plants in field conditions. Three minicircles derived from two infected plants, ranging in size from 1294 to 1503 nt (Table 1), were cloned and sequenced (MC#1, MC#2, and MC#3) (Fig. 1b). BLAST analysis revealed in all three minicircles 416- to 579-nt stretches with 95.2–100% identity to the reference BCTIV genome. These spanned the BCTIV intergenic region and included the conserved stem loop structure and the origin of viral rolling-circle replication[12] (Fig. 1b). The remaining sequences (of 735–1058 nt) differed between each minicircle but were all AT rich (69.8–71.2%) and lacked open reading frames. Interestingly, these sequences had no similarity to BCTIV or other known

geminiviruses or their known satellites. To test whether these sequences might derive from host chromosomal DNA, we performed PCR amplification on DNA from healthy *B. vulgaris* plants using primers specific for each minicircle insert. PCR products of predicted sizes were obtained (Supplementary Fig. 1a), indicating that these novel circular molecules are hybrids of viral and *B. vulgaris* chromosomal DNA. Therefore, they differ clearly from commonly observed satellite DNAs and defective viral molecules associated with begomovirus and mastrevirus infections[13,14], which do not include host-derived DNA.

To examine whether minicircles found in field-grown plants can also be generated under controlled infection conditions and whether they also arise in other BCTIV hosts, we delivered BCTIV to *B. vulgaris*, *Nicotiana benthamiana* and *Arabidopsis thaliana* plants using *Agrobacterium*-mediated inoculation (agroinfection)[15]. Four weeks post inoculation, DNA extracted from plants with BCTIV symptoms was subjected to RCA and enzymatic digestion. In addition to the complete BCTIV genome, experimentally infected plants also contained smaller circular DNA molecules of ~1300–1500 nt (Supplementary Fig. 1b). From *B. vulgaris* we cloned and sequenced two types of molecules: (a) hybrid minicircles consisting of viral and chromosomal DNA (Supplementary Fig. 1c) and (b) defective viral molecules consisting only of BCTIV sequence portions, with no host-derived DNA, whose genome organization and features are described in Supplementary Fig. 2a and Supplementary Table 1.

We examined in detail four different minicircles of 1337–1572 nt (MC#4 to MC#7) derived from three *B. vulgaris* plants (Supplementary Fig. 1c). Similar to those derived from field samples, these molecules were composed of portions of BCTIV, including the viral intergenic region (459–664 nt), linked to sequences with high AT content originating from the beet genome (820–998 nt) (Supplementary Fig. 1c and Table 1). Using primers specific for these non-viral portions of minicircles, we could again PCR-amplify chromosomal DNA fragments from healthy *B. vulgaris* plants (Supplementary Fig. 1a). Thus, minicircles are rapidly produced in experimental plants within 4 weeks of inoculation under controlled conditions.

Surprisingly, all small circular molecules formed after experimental infection of *N. benthamiana* and *A. thaliana* (Supplementary Fig. 1b) derived exclusively from BCTIV, thus representing defective BCTIV replicons (Supplementary Fig. 2b and 2c and Supplementary Table 1). Therefore, rapid and efficient formation of hybrid virus/host minicircles appears to be host-specific and thus may require host-derived factors; in contrast, defective viral molecules were produced in a wide range of BCTIV hosts.

Given that RCA may selectively detect amplified minicircles containing a specific restriction site and that subsequent cloning may favor DNA fragments present at high copy number, we also searched for minicircles using a genome-wide DNA next-generation sequencing (NGS) approach. Sequencing libraries were prepared with a PCR-free method using DNA isolated from BCTIV-infected *B. vulgaris* and *A. thaliana* plants, thus avoiding artifacts generated during DNA amplification. To identify minicircles, we developed a bioinformatics pipeline based on split-read mapping, followed by selection of reads with a 10-kb overlap around each viral/host junction. The recovered reads were de novo assembled through the a5_miseq pipeline and scaffolds including both viral and host genomes were selected (Fig. 2a). From a single BCTIV-infected *B. vulgaris* plant (plant 1 in Supplementary Fig. 1b), we retrieved eight scaffolds representing potential hybrid molecules with both viral and host genomic DNA. Among these, six scaffolds (scf#1, #2, #4, #5, #7 and #33) had two terminal virus/host junctions each and two scaffolds had only one virus/host junction (scf#56 and scf#59) (Fig. 2b).

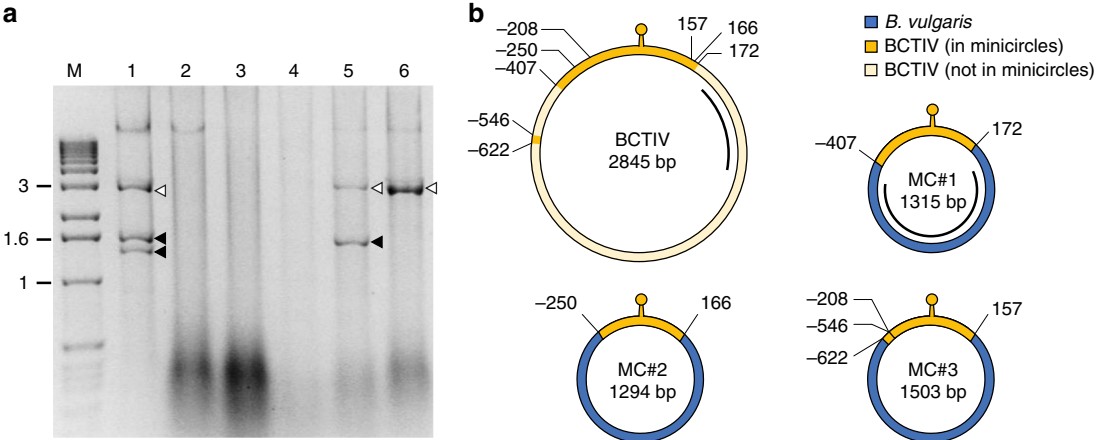

**Fig. 1** Identification of chimeric minicircles in field-grown sugar beet plants. **a** DNA fragments obtained after RCA and *Eco*RI digestion of DNA samples from different plants (lanes 1–6) collected from a field in Iran. The putative full-length BCTIV genome and smaller molecules are indicated by white and black arrows, respectively. Lane M, molecular markers with the size (kb) indicated on the left. Source data are provided as Supplementary Data 5. **b** Schematic representation of BCTIV and the three minicircles obtained from some of the BCTIV-infected field samples shown in **a**; MC#1 originates from plant 5, MC#2 and #3 from plant 1. The BCTIV and the minicircle genomes are represented in circular form; yellow and blue represent the BCTIV and the *B. vulgaris* genome-derived elements, respectively. The recombination coordinates on the BCTIV genome and the minicircles are indicated in nucleotides in relation to the conventional start of the BCTIV genome located at the stem-loop region (marked as a yellow lollipop). The locations of the MC#1- and BCTIV-specific probes used for subsequent southern blot analysis are represented by black lines. The complete minicircle sequences are reported in Supplementary Data 2

Notably, one of these scaffolds (scf #4) contained DNA sequence identical to that found in the MC#4 minicircle previously recovered by RCA from the same DNA plant sample (Fig. 2b). Altogether, the host DNA fragments found in scaffolds derived from various, apparently random, intergenic locations of the *B. vulgaris* genome (Supplementary Fig. 3a) with no evident association to precise chromosomal areas. However, they were again characterized by high AT content and lack of protein-coding potential (Table 1).

To examine whether the recovered scaffolds correspond to minicircles in vivo, we performed inverse PCR on DNA extracted from the infected plant using primers specific to the region of the minicircles derived from chromosomal DNA and sequenced the corresponding amplicons (Fig. 2c). This approach also revealed that the two scaffolds with only one virus/host junction (scf#56 and scf#59 in Fig. 2b) were parts of the same circular molecule (Supplementary Fig. 3b and Supplementary Data 1). As for minicircles obtained by RCA, all seven minicircles identified by NGS contained a viral intergenic region, with the stem loop and the origin of rolling-circle replication (Fig. 2d). Interestingly, although the contributions of the host and the virus-derived DNA varied in each minicircle, the length of the entire molecules seems to be limited to approximately half of the BCTIV genome, ranging from 1241 to 1572 nt (Fig. 2e). The presence in all minicircles of the viral origin of rolling-circle replication suggests that these molecules are amplified by the viral machinery, similarly to satellites and defective viral forms. Indeed, quantitative PCR assays directed towards host DNA captured in each minicircle showed significant increase in their copy number in the infected plants, with one minicircle having a copy number similar to the virus itself (compare scf#4 and C1 in Fig. 2f). Considering that from the same experimentally infected plant only one mini-circle was isolated by RCA (MC#4, identical to scf#4, Supplementary Fig. 1), whereas six additional ones were identified by NGS, detection by NGS appears more sensitive. Nonetheless, using the NGS strategy we did not identify hybrid scaffolds, and subsequently minicircles, in BCTIV-infected *A. thaliana* plants.

**Minicircles replicate and are transcribed in *B. vulgaris* plants and in other BCTIV hosts and are encapsidated into BCTIV virions.** To study the replication and systemic spread of hybrid minicircles during BCTIV infection, we generated an MC#1 construct to be delivered by agroinfection. *B. vulgaris* plants were agroinoculated with the MC#1 construct, either alone or together with the infectious clone of BCTIV. Using primers targeting a virus-derived region of MC#1 common to both MC#1 and BCTIV (Fig. 1b), systemic spread of MC#1 was detected by PCR only in plants co-inoculated with BCTIV, but not in plants inoculated with MC#1 alone (Supplementary Fig. 4a). Southern blot analysis with probes specific for the viral or the host portion of MC#1 confirmed the sizes and identities of the two replicons (Fig. 3a).

Next, we tested whether MC#1 minicircles can multiply and spread to other hosts of BCTIV, when co-infected with the supporting virus. For this, we performed analogous agroinfections on *A. thaliana* and *N. benthamiana* as hosts. As in *B. vulgaris*, MC#1 was amplified and spread systemically in both plant species only when co-inoculated with BCTIV (Fig. 3b, c). In all tested plants, MC#1 amplification did not significantly alter the disease symptoms induced by BCTIV (Supplementary Fig. 5a, b, c). Therefore, although spontaneous formation of minicircles seems to be restricted to BCTIV-infected *B. vulgaris*, their replication and systemic spread can also occur in *A. thaliana* and *N. benthamiana*, strictly dependent on co-infection with BCTIV.

Finally, as DNA encapsidation is a prerequisite for the systemic spread of monopartite geminiviruses, we tested directly whether MC#1 is incorporated into virions assembled by the BCTIV coat protein. Geminated virions from *N. benthamiana* plants co-inoculated with BCTIV and MC#1 were purified by sucrose gradient centrifugation (Supplementary Fig. 4b) and their BCTIV-specific protein composition confirmed by SDS-PAGE (Supplementary Fig. 4b, c). Importantly, nucleic acids extracted from purified virions included not only BCTIV DNA but also the *B. vulgaris* DNA present in MC#1 minicircle, as detected by southern blotting with BCTIV and MC#1-specific probes (Fig. 3d). Therefore, as MC#1 is encapsidated into BCTIV virions, it is very likely that minicircles can also spread to other

| Clone | Total length (nt) | BCTIV-derived sequence (nt) [%] | Beet-derived sequence (nt) | BCTIV sequences in minicircles[a] | AT % in BCTIV-derived sequence | AT % in beet-derived sequence | Locus of *B. vulgaris* (best BLAST hit) |
|---|---|---|---|---|---|---|---|
| **Table 1 Features of the minicircles cloned after RCA or identified by NGS from *B. vulgaris* (beet) plants infected by BCTIV** | | | | | | | |
| *RCA from field-grown samples* | | | | | | | |
| MC#1 | 1315 | 580 [44.1] | 735 | 1–172 … 2435–2845 | 61.4 | 69.8 | Beta chr6 9915362–9916096 |
| MC#2 | 1294 | 416 [32.0] | 878[b] | 1–166 … 2596–2845 | 61.3 | 71.2 | Beta chr7 36583873–36584676 |
| MC#3 | 1502 | 444 [29.3] | 1058 | 1–157 … 2223–2299 … 2637–2845 | 59 | 71.2 | Beta chr3:un.sca001 135440–136484 |
| *RCA from experimental infection* | | | | | | | |
| MC#4 | 1337 | 459 [34.33] | 878[b] | 1–135 … 1184–1234 … 2570–2845 | 62.3 | 73.6 | Beta chr6 57105781–57106674 |
| MC#5 | 1504 | 532 [35.4] | 972[b] | 1–267 … 2581–2845 | 62 | 70.6 | Beta chr3.sca001 1755016–1755723 |
| MC#6 | 1572 | 574 [36.5] | 998 | 1–188 … 2460–2845 | 61.5 | 68.2 | Beta chr5.sca008 17697504–17698497 |
| MC#7 | 1484 | 664 [44.7] | 820 | 1–327 … 2509–2845 | 61.1 | 71.2 | Beta chr1.sca002 10250861–10250043 |
| *NGS from experimental infection* | | | | | | | |
| scf #1 | 1407 | 396 [28.0] | 1011[b] | 1–110 … 2560–2845 | 60.36 | 71.97 | Beta chr4 23770925–23771936 |
| scf #2 | 1315 | 332 [25.24] | 983[b] | 1–177 … 2560–2845 | 60.25 | 73.25 | Beta chr8 1269745–1270665 |
| scf #4 | 1336 | 458 [34.28] | 878[b] | 1–135 … 1184–1234 … 2570–2845 | 62.3 | 73.6 | Beta chr6 57105781–57106674 |
| scf #5 | 1552 | 715 [46.1] | 837[b] | 1–196 … 2327–2845 | 59.31 | 71.45 | Beta chr8 33213612–33212757 |
| scf #7 | 1393 | 621 [44.6] | 772[b] | 1–329 … 2554–2845 | 61.20 | 73.06 | Beta chr6 470978–471718 |
| scf#33 | 1241 | 763 [61.48] | 478 | 1–108 … 2188–2845 | 56.49 | 73.44 | Beta 0118.sca00382 22639–23116 |
| scf#56/59 | 1429 | 453 [31.7] | 976 | 1–102 … 2495–2845 | 59.83 | 68.04 | Beta chr3 21716285–21716210 |

[a]BCTIV-Siv (JX082259) coordinates
[b]Incomplete and/or discontinuous matching

hosts by the insect vector propagating BCTIV itself, similarly to defective DNAs[16].

Bearing in mind that the *B. vulgaris* sequence found in minicircles is always positioned downstream of the intergenic region of BCTIV, which contains the promoter for the viral sense transcript, we tested whether host DNA captured by the virus is transcribed during MC#1 replication. Using RT-PCR, we detected a novel *B. vulgaris*-specific transcript encompassing the captured portion of MC#1 in both *B. vulgaris* and *N. benthamiana* plants in which both MC#1 and BCTIV were replicating (Fig. 3e).

## Discussion

It is evident that the sizes of minicircles, their rapid and efficient formation, and the dependence of replication on viral infection, transcription and trans-encapsidation parallel defective DNAs and viral satellites[16–18]. It is interesting that, like minicircles, satellites associated with geminiviruses are also characterized by a high AT content and homology to helper viruses that is limited to a conserved region, including the stem-loop sequences[14]. Thus, the generation of satellites may be initiated by the formation of minicircles.

Previous studies of insect DNA insertions into baculoviruses associated this process with the mobilization of transposons that results in insertions into the virus[4,19,20]. However, as fragments of *B. vulgaris* DNA found in minicircles cannot be assigned to any known transposons, it is unlikely that transposition contributes to the formation of these molecules. However, the apparent absence

of minicircles in BCTIV-infected *A. thaliana* and *N. benthamiana* observed here and following infection of *B.vulgaris* with unrelated geminiviruses[17] suggests that only a specific combination of host and virus activities results in conditions promoting initial minicircle formation. It is then to be expected that such specific host/virus compatibility influences the direction of the flow of captured host DNA within populations of infected plants.

BCTIV is transmitted by the leafhopper *Circulifer haematoceps* in a circulative non-propagative manner, where the virus circulates through the insect body and accumulates in the salivary glands[21]. Due to the polyphagous behavior of this vector and the wide host range of BCTIV[11,22–24], encapsidated minicircles could be exported over a wide range of plants, similarly to defective DNAs[16].

Remarkably, circular DNA molecules, referred to as endogenous viral elements, are considered as precursors of integration of viral sequences into host genomes[25–27]. Although the integration of geminiviruses during infection in real-time was never observed, geminiviruses-related DNA sequences are frequently found in plant genomes, indicating historical integration events[28–32]. In theory, such events of integration in host genome should occur also for minicircles.

Different mechanisms have been proposed as potential mediator of HGT in plants[33], involving large and well characterized DNA sequences, such as ribosomal repeats[34], or genes with bacterial or fungal origin[35,36]. However, considering that minicircles consist mostly of variable non-viral DNA, their integration into plant genomes would be difficult to recognize using standard

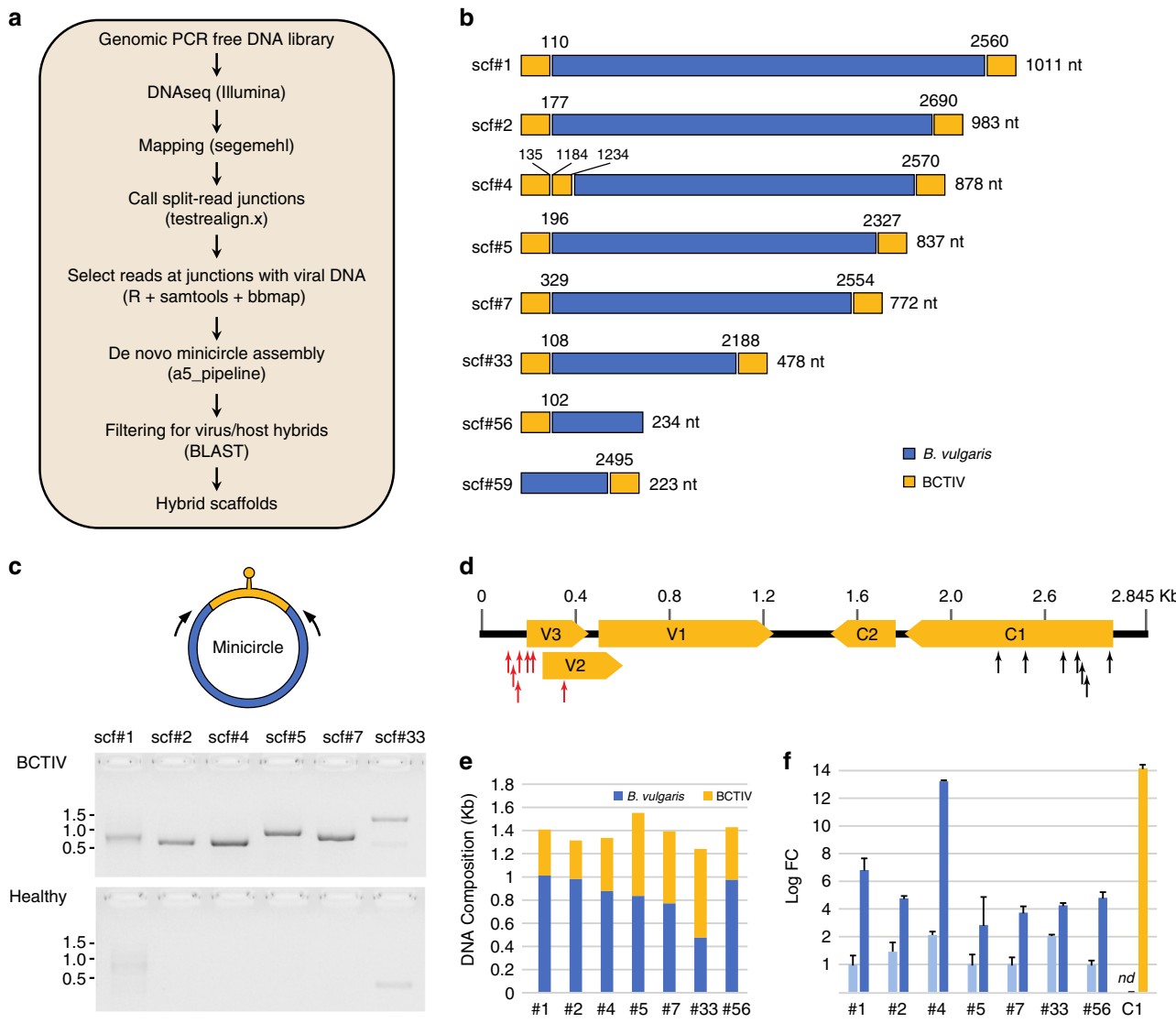

**Fig. 2** Identification of minicircles in experimentally infected *Beta vulgaris* plants by the NGS approach. **a** Diagram of the bioinformatic pipeline used for the detection of minicircles. The tools used in each step are indicated in parentheses (refer to the Methods section for details). **b** Schematic representation of the eight BCTIV/*Beta vulgaris* hybrid scaffolds assembled after the filtering step described in **a**. The junction coordinates referring to the BCTIV genome are indicated above each recombination point. The sizes (nt) of the *B. vulgaris* DNA fragments (blue portions) are reported on the right. **c** In vivo validation of the circular nature of the scaffolds reported in **b**, performed by inverse PCR (see also Supplementary Figure 3). The approximate positions of the primers specific for the non-viral portions of each minicircle (in blue) are indicated by arrows. DNA from mock-inoculated *B. vulgaris* was used as a negative control. Molecular markers (kb) are shown on the side. Source data are provided as Supplementary Data 5. **d** Distribution of minicircle junctions between viral and host DNA. The proximal and distal junctions are marked with red and black arrows, respectively. BCTIV ORFs are represented by yellow boxes. **e** Relative contributions of *B. vulgaris* and BCTIV DNA to minicircles identified by NGS. **f** Relative increase in copy number of chromosomal DNA fragments due to their amplification as parts of minicircles in the BCTIV-infected plant. Copy number was determined by quantitative PCR and normalized to a single copy gene (GAPDH). Light-blue bars represent healthy plants and dark-blue bars BCTIV-infected plants. BCTIV copy number was estimated by amplification of a fragment of the C1 ORF (yellow bar). Each bar represents the mean of three repetitions; ±s.d. marked by error bars; nd not detected. Source data are provided as Supplementary Data 5

genomic annotation studies. Therefore, genetic consequences of horizontal transfer of host DNA during viral infection by the mechanism described here are likely to be overlooked.

## Methods

**Plant material and nucleic acid extraction**. Field-grown *B. vulgaris* plants were collected in the Fars province of Iran in 2008. *N. benthamiana* and *B. vulgaris* L. var. *vulgaris* plants (spinach beet) were grown in soil in a greenhouse at 20–28/16–20 °C (day/night) and a 16/8 h (light/dark) photoperiod, with supplementary lighting when necessary. *A. thaliana* Col-0 plants were grown in a mixture of soil and perlite (1:1) in a growth chamber at 21 °C with a 12/12 h (light/dark)

photoperiod. Total nucleic acids were extracted from plant samples (100 mg) using the TLES method[37].

**Rolling circle amplification (RCA) and cloning of minicircle DNA molecules**. Nucleic acids were subjected to RCA using the TempliPhi Kit (GE Healthcare, Little Chalfont, UK) according to Soleimani et al. (2013). After digestion of the RCA product with either *Eco*RI or *Apa*I (both having single restriction sites in the BCTIV genome), DNA was electrophoretically separated on 1% agarose gels in 0.5X TBE buffer. Fragments of 1.3–1.5 kb were collected after ethidium bromide staining, eluted with the High Pure PCR Product Purification Kit (Roche Diagnostics, Indianapolis, Indiana, USA) and cloned into pBluescript KS+ linearized with the appropriate enzymes. Each insert was sequenced on both strands by an

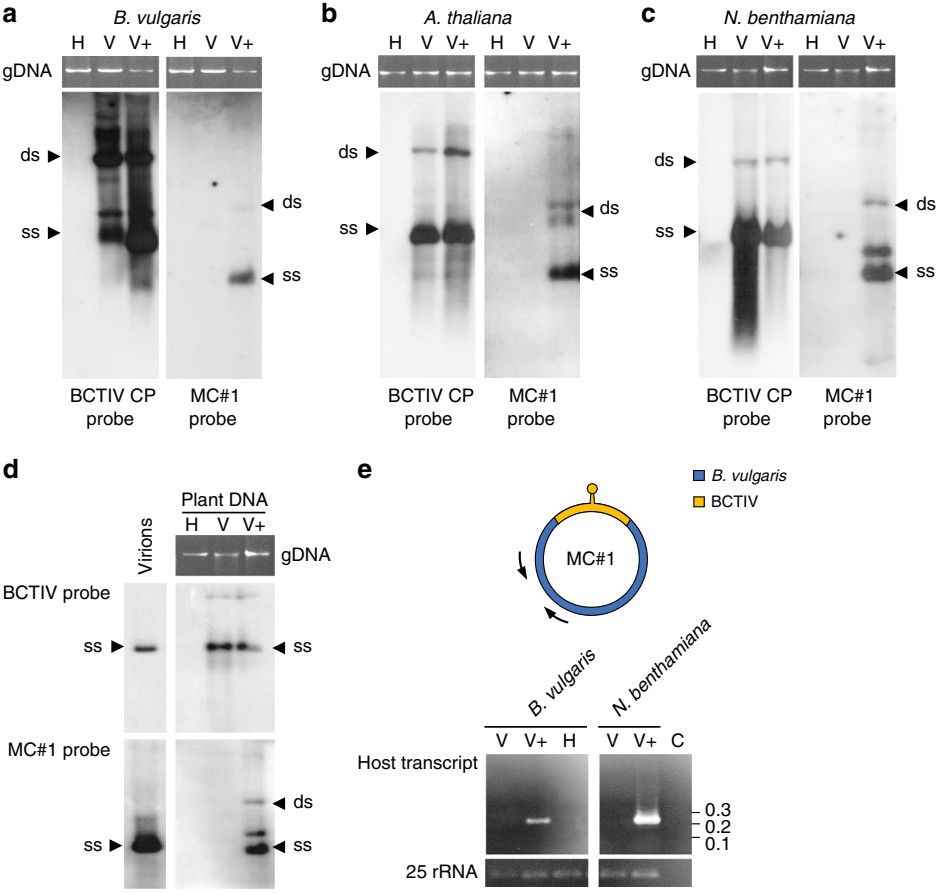

**Fig. 3** MC#1 replication and transcription when transfected to various hosts. Southern blots of DNA extracted from *B. vulgaris* (**a**), *A. thaliana* (**b**) and *N. benthamiana* (**c**), mock-inoculated healthy plants (H); plants agroinfected with BCTIV (V) or with both BCTIV and MC#1 (V+). Leaves were collected at 4 weeks post inoculation (wpi) and hybridized with BCTIV- or MC#1-specific probes (see Fig. 1b). Bands of single and double stranded DNAs are indicated as "ss" and "ds", respectively. Plant genomic DNA (gDNA) is shown above the blots as a loading control. Source data are provided as Supplementary Data 5. **d** Southern blot of DNA extracted from purified BCTIV virions and from the *N. benthamiana* plants used for purification of BCTIV particles, hybridized with the BCTIV- and the MC#1-specific probes (see Fig. 1b). "ss", "ds" and gDNA as in **a–c**. Source data are provided as Supplementary Data 5. **e** Transcription of the beet-derived DNA sequence of MC#1 in *B. vulgaris* and *N. benthamiana* plants inoculated with BCTIV and MC#1 (V+) detected by RT-PCR with primers positioned as indicated by arrows in the upper part of the panel. Lanes are marked H, V and V+ as in **a–d**, whereas **C** is a RT-PCR-negative reaction control. Source data are provided as Supplementary Data 5

automatic sequencing service (BMR, Padua, Italy). The sequences of the identified minicircles are reported in Supplementary Data 2.

**Construction of an infectious clone of the MC#1 minicircle.** The full-length clone of MC#1 (accession no. JX082260), named pBS-1.0mer-MC#1, was used to obtain a head-to-tail 1.4-mer clone. For this, a 545-nt fragment was excised from pBS-1.0mer-MC#1 by digestion with *Apa*I and re-ligation, resulting in pBS-0.4mer-MC#1. The full-length *Eco*RI insert from pBS-1.0mer-MC#1 was cloned into pBS-0.4mer-MC#1 to produce pBS-1.4mer-MC#1. The 1.4-mer sequence was excised with *Pvu*II, ligated into the binary vector pBin19 and linearized with *Sma*I to create pBin1.4mer-MC#1. This plasmid was transferred to *Agrobacterium tumefaciens* LBA4404 by the freeze and thaw method for subsequent agroinfection.

**Plant inoculation.** Plants were agroinoculated with *A. tumefaciens* carrying the 1.4mer BCTIV-Siv clone[11] (accession no. JX082259), pBin1.4mer-MC#1 or pBIN19 as control, in the appropriate combinations. Virus or minicircle replication was assessed by PCR amplification or molecular hybridization on DNA extracts prepared using the Dot-Blot method[38].

**PCR analysis.** The nature of the non-viral DNA portions of minicircles was examined with primers listed in Supplementary Table 2. For cloned minicircles (MC#1 to MC#7), primers were designed on the non-viral DNA portions to validate their host plant origin (Fig. 1e).

**Southern blot hybridization.** Total DNA (~400 ng) was separated on 1% agarose gels in 0.5X TBE buffer containing 0.5 μg/ml ethidium bromide. The bands were capillary transferred to positively charged nylon membranes (Roche Diagnostics) and UV-crosslinked. Blots were hybridized with digoxygenin-labeled probes obtained by incorporating DIG-dUTP with the PCR DIG Probe Synthesis Kit (Roche Diagnostics). Probes specific for either the virus or the cloned MC#1 molecule consisted of part of the BCTIV Coat Protein gene amplified with the BCTIRV-F and BCTIRV-R primers, or a portion of the plant-derived sequence of MC#1 amplified with the primers MC#1_682F and MC#1_925R (Supplementary Table 2).

**Virus purification.** BCTIV particles were extracted from infected *N. benthamiana* leaves at 3 weeks post inoculation (wpi) using a protocol originally described for *Digitaria streak virus*[39]. Briefly, the frozen tissue was ground in extraction buffer (100 mM K-phosphate buffer pH 7.0, 20 mM Na$_2$SO$_3$, 10 mM EDTA, and 10 mM diethyl-dithiocarbammate, at a ratio of 0.35 g/ml buffer). The homogenate was filtered through a nylon mesh and Triton X-100 was added to a final concentration of 0.1%. After stirring at 4 °C for 1 h, a chloroform:butanol (1:1) mixture was added to the homogenate (1/10 vol). After further stirring for 20 min, the extract was centrifuged for 10 min at 7600×g (SS-34 rotor; Thermo Fisher Scientific, Langenselbold, Germany) at 4 °C. PEG$_{8000}$ and NaCl were added to the aqueous phase at final concentrations of 12% and 0.2 M, respectively. Samples were stirred for 2 h at 4 °C and centrifuged (as above). The pellet was dissolved in 0.1 M K-phosphate buffer pH 7.0 with 10 mM EDTA (1/10 of the original volume) and stirred overnight at 4 °C. After spinning (as above), the supernatant was centrifuged for 90 min at 240,000×g (Beckman 55.2 Ti rotor; Beckman, CA) and the pellet resuspended in

0.1 M K-phosphate buffer pH 7.0 with 10 mM EDTA. The suspension was loaded onto a 40–10 % sucrose gradient prepared in 0.1 M K-phosphate buffer pH 7.0 with 10 mM EDTA and centrifuged for 90 min at 178,000×g (Beckman SW 41Ti rotor). Fractions were collected and spun for 30 min at 390,000×g (Beckman TL100 rotor) and the pellet then resuspended in 100 µl of 0.1 M K-phosphate buffer pH 7.0. DNA was extracted from an aliquot of this suspension[37] for southern blot hybridization, whereas another aliquot was dissolved in 3X Laemmli sample buffer and boiled at 100 °C for 2 min to solubilize proteins.

**Protein analysis**. Protein extracts were separated on 4–20% acrylamide gradient mini-protean TGX gels (Bio-Rad, Richmond, CA) in Tris-glycine buffer and stained with Coomassie brilliant blue according to standard procedures.

**Electron microscopy examination**. Aliquots of the above preparations were adsorbed onto carbon-coated grids for ~1–3 min, removing excess fluid with filter paper. The grids were negatively stained with aqueous 0.5% uranyl acetate, observed and photographed using a CM 10 electron microscope (Philips, Eindhoven, The Netherlands).

**Genomic library preparation**. Before library preparation, genomic DNA was fragmented to an average insert size of 350 nt using 18 cycles of 30 s each on a Bioruptor Diagenode sonication device, split into three groups of 6 cycles with 2.5 min on ice between each set. The genomic DNA sequencing libraries were prepared using the TruSeq DNA PCR-Free LT Library Prep Kit following the manufacturer's instructions (Illumina, San Diego, CA), starting from 1.1 µg of sonicated DNA. Libraries were validated using High Sensitivity D1000 screentape on the 2200 Tapestation instrument (Agilent technologies, Santa Clara, CA) and the Light-Cycler 480 Instrument II using the LightCycler 480 SYBR Green I Master mix (Roche, Basel, Switzerland).

**Minicircle detection pipeline**. The DNA libraries were sequenced with 2 × 76-bp paired-end reads on an Illumina NextSeq 500. The raw reads were trimmed using Trimmomatic to remove adapter sequences. Reads with an averaged value of at least 15 in a 4-nt window were trimmed from both ends. After trimming, reads pairs with at least one mate shorter than 16 bp were discarded. The remaining sequences (on average 94% of raw reads) were aligned with segemehl v0.2.0[40], with a 95% accuracy threshold, to the corresponding reference genome (A. thaliana TAIR10 and the B. vulgaris RefBeet-1.2.2), including also the sequence of the BCTIV genome. Split read junctions were called with testrealign.x (part of the segemehl toolkit) with default parameters, and junctions with a minimum coverage of 10 reads and comprising the virus genome were selected for downstream analysis using R (dplyr package). The reads spanning the 10-kb locus around each selected junction were then recovered, together with their mates from the original fastq files, using SAMtools[41] and BBMap (https://sourceforge.net/projects/bbmap/). The filtered paired read lists were entered into the A5_miseq pipeline[42] with default parameters for reference-unbiased de novo assembly. Then, blastn (BLAST v2.2.25+) with default parameters was applied to filter only assembled scaffolds matching both the reference and the BCTIV genomes in order to select minicircle candidates. The general metrics of DNA alignment analysis are shown in Supplementary Table 3. The sequences of the filtered scaffolds from this analysis are shown in Supplementary Data 3.

**Sequence analysis**. Comparisons with sequences available in the databases were carried out using BLAST (www.ncbi.nlm.nih.gov) and ENSEMBL (http://plants.ensembl.org/index.html) tools. The original chromosomal locations of the scaffold sequences found in minicircles were identified using blastn (BLAST v2.2.25+) with a database build with the B. vulgaris (RefBeet-1.2.2) genome assembly (list of all blast hits is provided as Supplementary Data 4). Multiple sequence alignments and identification of ORFs were performed with CLUSTAL V (DNAStar MegAlign software). GC rich regions were sought using DNA compositions base tools (http://www.endmemo.com/bio/gc.php). The presence of tandem repeats was verified using the Tandem Repeat Finder program (https://tandem.bu.edu/trf/trf.html).

**PCR validation of minicircle amplification**. The seven minicircle candidates found by NGS in B. vulgaris were validated by PCR with inverted primers pairs (Supplementary Table 2) designed on the plant genome portions in order to amplify the junctions between plant and virus DNA sequences. The DNA fragments obtained were cloned into pGEM-T Easy (Promega) and sequenced to validate the in silico predicted recombination junctions (Supplementary Data 1). Quantitative PCR was carried out in triplicate using 10 ng of template genomic DNA, 200 nM target-specific primers (Supplementary Table 2) and LightCycler 480 SYBR Green I Master (Roche) in the LightCycler 480 II detection system (Roche) in a volume of 10 µl.

## Data availability

Sequencing data have been deposited in Gene Expression Omnibus under the accession number GSE114958. The sequence of MC#1 is deposited in GenBank under the accession number JX082260. A reporting summary for this Article is

available as a Supplementary Information file. The source data underlying Figs. 1a, 2c, f, 3a–e, and Supplementary Figs 1a, b, 3b, and 4a–c are provided as a Supplementary Data 5. All the other data generated or analyzed during this study are included in this published article or available under request.

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

## Acknowledgements

The authors thank Marta Vallino for electron microscopy and Laura Miozzi for help in bioinformatic analysis. This work was supported by the European Research Council (EVOBREED) [322621] and a Gatsby Fellowship [AT3273/GLE].

## Author contributions

M.C., G.P.A. and E.N. designed the experiments; S.A.A.B. provided biological samples, T. J. prepared nucleic acid libraries; M.C., E.N., A.M.V., S.M., N.V., T.J. and R.S. performed the experiments, M.C. analyzed genomic sequencing data; J.P. contributed reagents/ materials/analysis tools. M.C., E.N. and J.P. wrote the paper.

## Additional information

**Competing interests:** The authors declare no competing interests.

