## [Peer Review File · Nature Communications]

Reviewers' comments:

Reviewer #1 (Remarks to the Author):

Overall, I found these results to be quite interesting. I did have some minor issues, as detailed below, but I think these results are well worth publishing. I also liked the fact that the authors did not have the need to provide massive amounts of peripheral supplemental data. One quibble is the use of the phrase "virus-mediated horizontal transfer of chromosomal DNA between plant species." because this normally implies movement from the genome of one plant into the genome of another. Unless these hybrid circles are integrated into the genome of the target plant, this really isn't the case. Further, HGT also generally refers to transfer to the germline, not to somatic cells. Finally, it would have been nice if some additional information concerning the transferred DNA. Are they really just random AT-rich sequences? Is it informative that they appear to all be single copy?

Line 54. Were these healthy plants from Iran, or those grown in the laboratory? Was a negative control (viral primers) also used?

Line 57. "They differ clearly from commonly observed..." How so?

Line 70. This section, along with Figure S2 should be rewritten for clarity.

Line 102. No preference for euchromatin versus heterochromatin? Genic versus intergenic? Expressed versus not expressed? Since the data is available, why not do a bit more analysis?

Line 115. "that the viral machinery amplifies these similar to satellites" awkward wording.

Line 127. This is confusing to me. Wouldn't you want to use primer pairs specific to either the construct or the virus so that you could distinguish them? That being said, your Southern blots are convincing.

Figure 1. Where is the negative control here?

Figure 2C. Where is the negative control here?

Reviewer #2 (Remarks to the Author):

This paper reports the presence of hybrid host-virus DNA molecules encapsidated into particles of the Beet curly top Iran virus that replicated into *Beta vulgaris*. The authors further show that the virus efficiently replicates these hybrid molecules which are transcribed and can spread systematically in *B. vulgaris* but also in two other plant species.

The paper is well written, the experiments seem to have been conducted in a thorough way and in my opinion the results are robust and they do support the interpretations of the authors. I only have minor comments.

1 – I very much like the concise nature of the paper but in some aspects I find it too concise. I encourage the authors to discuss in more details their results in the context of the literature on horizontal transfer and on how we think a HT event may occur in plants. What is the tissue tropism of BCTIV? How likely is it that the hybrid host-DNA molecules be integrated into the germline genome of another plant? What mechanisms could underlie such integration (DNA-based? RNA-based? Both?)? Are there examples of endogenous geminiviruses in the literature?

2 – A few more informations on the general biology and ecology of geminiviruses could also help better assess the likelihood with which these viruses may serve as vectors of HT in plants. How widespread is BCTIV? What is its mode of transmission? Is it frequently horizontally transmitted between plants?

3 – I also encourage the authors to further discuss their results in light of what is known on HGT in plants (e.g. Yue et al. 2012 Nat. Commun. ; Quispe-Huamanquispe et al. 2017 Front Plant Sci. ; Gao et al. 2014 Funct. Integr. Genomics; Mahelka et al. 2017 PNAS and other). For instance, do they think that geminiviral-mediated HT can explain the HGT events described in Mahelka et al. 2017?

4 – In previous publications describing insect to virus HT of TEs, some of the TEs were shown to have been horizontally transferred in the past between several insect species. Have the authors looked

for evidence of between-plants HT of the host DNA sequences found in hybrid host/virus molecules?
This could be done by searching all plant genomes using blastn.

5 – Another point that could be touched in the discussion relates to the potential costs associated with the replication of such hybrid molecules for the virus. I assume this has already been studied and discussed elsewhere in the context of defective DNAs and viral satellites? Could the formation of host-virus hybrid molecules allow the host population to better tolerate or better resist to the virus? It is shown here that the symptoms of the virus do not differ whether hybrid molecules are present or absent, but could it be possible that such molecules decrease the efficiency of plant-to-plant transmission of the virus?

Reviewers' comments:

Reviewer #1 (Remarks to the Author):

Overall, I found these results to be quite interesting. I did have some minor issues, as detailed below, but I think these results are well worth publishing. I also liked the fact that the authors did not have the need to provide massive amounts of peripheral supplemental data. One quibble is the use of the phrase “virus-mediated horizontal transfer of chromosomal DNA between plant species.” because this normally implies movement from the genome of one plant into the genome of another. Unless these hybrid circles are integrated into the genome of the target plant, this really isn't the case. Further, HGT also generally refers to transfer to the germline, not to somatic cells.

We agree with the reviewer on the fact that we did not observe real-time integration of minicircles in the host genome. In the revised version of our manuscript we corrected the sentence lowering our claim. Now it reads (line 12):

“Thus, we have likely documented in real time the initial steps of a possible path of virus-mediated horizontal transfer of chromosomal DNA between plant species”

It is commonly accepted that circular DNA molecules are a precursor of integration in the host genome, and there are many evidences of Geminivirus elements integrated in the plant genome, indicating that past events of stable insertion occurred in the germline (we now discuss this concept in a new paragraph at lines 177-182). We therefore think that it is very plausible that minicircles will not represent an exception and, similarly to other viral-derived circular DNAs, will possibly integrate in the host genome in evolutionary time.

Finally, it would have been nice if some additional information concerning the transferred DNA. Are they really just random AT-rich sequences? Is it informative that they appear to all be single copy?

Unfortunately, additional patterns are not emerging from our data, beside what already described in the manuscript. The fact that for most of the transferred DNA a single top scored blast hit was identified does not implicate that the DNA sequence is single copy in the beta genome. In fact, some sequences scored several significant hits, suggesting that they are repeated in the genome. We now included a better explanation of this analysis (lines 300-303) and, as Supplementary Data 4, the list of all blast hits identified.

Line 54. Were these healthy plants from Iran, or those grown in the laboratory? Was a negative control (viral primers) also used?

The healthy plants used in this analysis were grown in the laboratory, in pathogen-free controlled conditions. No viral primers were used in a parallel PCR. However, plants from the same seed batch were used as healthy negative controls in the Southern blots of Figure 3, without reacting with the virus-specific probe. Therefore, they can be considered healthy.

Line 57. “They differ clearly from commonly observed...” How so?

The sentence (line 53) was changed accordingly, now it reads:

“Therefore, they differ clearly from commonly observed satellite DNAs and defective viral molecules associated with begomovirus and mastrevirus infections (Xhou, 2013; Kumar et al., 2014), which do not include host-derived DNA”.

Line 70. This section, along with Figure S2 should be rewritten for clarity.

The section (lines 63-67) has been rewritten and now it reads:

“From *B. vulgaris* plants we cloned and sequenced two types of molecules: (a) hybrid minicircles consisting of viral and chromosomal DNA (Supplementary Fig. 1c) and (b) defective viral molecules consisting only of BCTIV sequence portions, with no host-derived DNA, whose genome organization and features are described in Supplementary Fig. 2a and Supplementary Table 1.”

Line 102. No preference for euchromatin versus heterochromatin? Genic versus intergenic? Expressed versus not expressed? Since the data is available, why not do a bit more analysis?

All sequences identified have low coding potential and they derived from intergenic regions, we modified the relevant sentence of the manuscript including additional information, at lines 98-101. Now it reads:

“Altogether, the host DNA fragments found in scaffolds derived from various, apparently random, intergenic locations of the *B. vulgaris* genome (Supplementary Fig. 3a) with no evident association to precise chromosomal areas. However, they were again characterized by high AT content and lack of protein-coding potential (Table 1).”

Line 115. “that the viral machinery amplifies these similar to satellites” awkward wording.

The sentence (at line 112) now reads:

“The presence in all minicircles of the viral origin of rolling-circle replication suggests that these molecules are amplified by the viral machinery, similarly to satellites and defective viral forms”

Line 127. This is confusing to me. Wouldn't you want to use primer pairs specific to either the construct or the virus so that you could distinguish them? That being said, your Southern blots are convincing.

Considering that minicircles are smaller than the BCTIV genome, the PCR assay was designed to screen plants with a single primer pair, as specified in the text. As commented by the reviewer, the Southern blot results made further amplification with specific primer pairs unnecessary.

Figure 1. Where is the negative control here?

Figure 1 refers to field collected samples. In such conditions, it is impossible to control which plants have been exposed to the pathogen and their stage of infection. Therefore, the collection of a proper negative control is not possible.

Figure 2C. Where is the negative control here?

Negative controls (healthy plants) are reported in the lower panel of the figure. If reviewer refers to the blank PCR reactions (with no DNA added), these are now available in the provided source data file.

Reviewer #2 (Remarks to the Author):

This paper reports the presence of hybrid host-virus DNA molecules encapsidated into particles of the Beet curly top Iran virus that replicated into *Beta vulgaris*. The authors further show that the virus efficiently replicates these hybrid molecules which are transcribed and can spread systematically in *B. vulgaris* but also in two other plant species.

The paper is well written, the experiments seem to have been conducted in a thorough way and in my opinion the results are robust and they do support the interpretations of the authors. I only have minor comments.

1 – I very much like the concise nature of the paper but in some aspects I find it too concise. I encourage the authors to discuss in more details their results in the context of the literature on horizontal transfer and on how we think a HT event may occur in plants. What is the tissue tropism of BCTIV? How likely is it that the hybrid host-DNA molecules be integrated into the germline genome of another plant? What mechanisms could underlie such integration (DNA-based? RNA-based? Both?)? Are there examples of endogenous geminiviruses in the literature?

To answer this point and to provide some additional elements of discussion, we added a new section to the discussion (lines 177-182). It reads:

“Remarkably, circular DNA molecules, referred to as endogenous viral elements, are considered as precursors of integration of viral sequences into host genomes (Crochu et al., 2004; Palatini et al., 2017; Suzuki et al., 2017). Although the integration of geminiviruses during infection in real-time was never observed, geminiviruses-related DNA sequences are frequently found in plant genomes, indicating historical integration events (Ashby et al., 1997; Bejarano et al., 1996; Filloux et al., 2015; Kenton et al., 1995; Murad et al., 2004). In theory, such events of integration in host genome should occur also for minicircles.”

2 – A few more informations on the general biology and ecology of geminiviruses could also help better assess the likelihood with which these viruses may serve as vectors of HT in plants. How widespread is BCTIV? What is its mode of transmission? Is it frequently horizontally transmitted between plants?

We discussed this topic in a new section (lines 172-176), it reads:

“BCTIV is transmitted by the leafhopper *Circulifer haematoceps* in a circulative non-propagative manner, where the virus circulates through the insect body and accumulates in the salivary glands (Soto and Gilbertson, 2003). Due to the polyphagous behavior of this vector and the wide host range of BCTIV (Anabestani et al., 2017; Eini et al., 2016; Gharouni Kardani et al., 2013; Soleimani et al., 2013), encapsidated minicircles could be exported over a wide range of plants, similarly to defective DNAs (Patil and Dasgupta, 2006)”.

3 – I also encourage the authors to further discuss their results in light of what is known on HGT in plants (e.g. Yue et al. 2012 Nat. Commun. ; Quispe-Huamanquispe et al. 2017 Front Plant Sci. ; Gao

et al. 2014 *Funct. Integr. Genomics*; Mahelka et al. 2017 *PNAS* and other). For instance, do they think that geminiviral-mediated HT can explain the HGT events described in Mahelka et al. 2017?

Due to the size constraint of minicircles and the high AT content, putative geminiviral-mediated HT events should not be easily detected in genome comparison studies, and they are not consistent with the large rDNA repeats transfers observed by Mahelka and collaborators.

To extend our discussion and fulfil the reviewer request, we add a new section to our manuscript at lines 183-189, it reads:

“Different mechanisms have been proposed as potential mediator of HGT in plants (Gao et al., 2014), involving large and well characterized DNA sequences, such as ribosomal repeats (Mahelka et al., 2017), or genes with bacterial or fungal origin (Quispe-Huamanquispe et al., 2017; Yue et al., 2012). However, considering that minicircles consist mostly of variable non-viral DNA, their integration into plant genomes would be difficult to recognize using standard genomic annotation studies. Therefore, genetic consequences of horizontal transfer of host DNA during viral infection by the mechanism described here are likely to be overlooked.”

4 – In previous publications describing insect to virus HT of TEs, some of the TEs were shown to have been horizontally transferred in the past between several insect species. Have the authors looked for evidence of between-plants HT of the host DNA sequences found in hybrid host/virus molecules? This could be done by searching all plant genomes using blastn.

The properties of the DNA sequences integrated in minicircles are not compatible with active TEs, and BLAST does not report any similarity with known transposons or transposon factors. At this stage, we simply cannot find any evidence of TEs involved in minicircle formation.

5 – Another point that could be touched in the discussion relates to the potential costs associated with the replication of such hybrid molecules for the virus. I assume this has already been studied and discussed elsewhere in the context of defective DNAs and viral satellites? Could the formation of host-virus hybrid molecules allow the host population to better tolerate or better resist to the virus? It is shown here that the symptoms of the virus do not differ whether hybrid molecules are present or absent, but could it be possible that such molecules decrease the efficiency of plant-to-plant transmission of the virus?

The investigation of the potential effects of minicircles in the plant-to-plant transmission is a very interesting topic, requiring different experimental work and not in the scope of the current manuscript. We therefore decided to focus our discussion on the most evidence-supported conclusions.

REVIEWERS' COMMENTS:

Reviewer #1 (Remarks to the Author):

The authors have satisfied my concerns.